# Generating Natural Adversarial Examples

**Zhengli Zhao**
University of California
Irvine, CA 92697, USA
`zhengliz@uci.edu`

**Dheeru Dua**
University of California
Irvine, CA 92697, USA
`ddua@uci.edu`

**Sameer Singh**
University of California
Irvine, CA 92697, USA
`sameer@uci.edu`

## Abstract

Due to their complex nature, it is hard to characterize the ways in which machine learning models can misbehave or be exploited when deployed. Recent work on adversarial examples, i.e. inputs with minor perturbations that result in substantially different model predictions, is helpful in evaluating the robustness of these models by exposing the adversarial scenarios where they fail. However, these malicious perturbations are often unnatural, not semantically meaningful, and not applicable to complicated domains such as language. In this paper, we propose a framework to generate natural and legible adversarial examples that lie on the data manifold, by searching in semantic space of dense and continuous data representation, utilizing the recent advances in generative adversarial networks. We present generated adversaries to demonstrate the potential of the proposed approach for black-box classifiers for a wide range of applications such as image classification, textual entailment, and machine translation. We include experiments to show that the generated adversaries are natural, legible to humans, and useful in evaluating and analyzing black-box classifiers.

## 1 Introduction

With the impressive success and extensive use of machine learning models in various security-sensitive applications, it has become crucial to study vulnerabilities in these systems. Dalvi et al. (2004) show that adversarial manipulations of input data often result in incorrect predictions from classifiers. This raises serious concerns regarding the security and integrity of existing machine learning algorithms, especially when even state-of-the-art models including deep neural networks have been shown to be highly vulnerable to adversarial attacks with intentionally worst-case perturbations to the input (Szegedy et al., 2014; Goodfellow et al., 2015; Kurakin et al., 2016; Papernot et al., 2016b; Kurakin et al., 2017). These adversaries are generated effectively with access to the gradients of target models, resulting in much higher successful attack rates than data perturbed by random noise of even larger magnitude. Further, training models by including such adversaries can provide machine learning models with additional regularization benefits (Goodfellow et al., 2015).

Although these adversarial examples expose "blind spots" in machine learning models, they are *unnatural*, i.e. these worst-case perturbed instances are not ones the classifier is likely to face when deployed. Due to this, it is difficult to gain helpful insights into the fundamental decision behavior inside the black-box classifier: why is the decision different for the adversary, what can we change in order to prevent this behavior, and is the classifier robust to natural variations in the data when not in an adversarial scenario? Moreover, there is often a mismatch between the input space and the *semantic space* that we can understand. Changes to the input we may not think meaningful, like slight rotation or translation in images, often lead to substantial differences in the input instance. For example, Pei et al. (2017) show that minimal changes in the lighting conditions can fool automated-driving systems, a behavior adversarial examples are unable to discover. Due to the unnatural perturbations, these approaches cannot be applied to complex domains such as language, in which enforcing grammar and semantic similarity is difficult when perturbing instances. Therefore, existing approaches that find adversarial examples for text often result in ungrammatical sentences, as in the examples generated by Li et al. (2016), or require manual intervention, as in Jia & Liang (2017).

In this paper, we introduce a framework to generate *natural* adversarial examples, i.e. instances that are meaningfully similar, valid/legible, and helpful for interpretation. The primary intuition

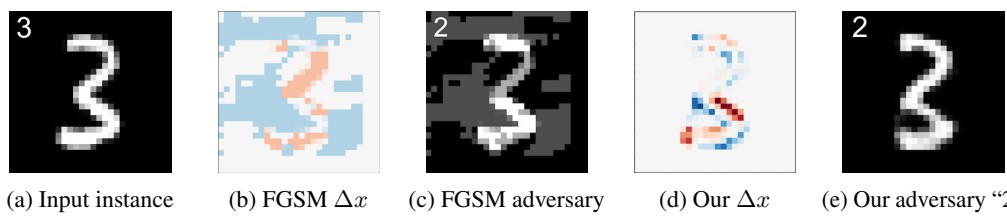

(a) Input instance    (b) FGSM $\Delta x$    (c) FGSM adversary    (d) Our $\Delta x$    (e) Our adversary "2"

Figure 1: **Adversarial examples.** Given an instance (a), existing FGSM approach (Goodfellow et al., 2015) adds small perturbations in (b), that change the prediction of the model (to be "2", in this case). Instead of such random-looking noise, our framework generates natural adversarial examples, such as in (e), where the differences, shown in (d) (with blue/+, red/-), are meaningful changes to the strokes.

behind our proposed approach is to perform the search for adversaries in a dense and continuous representation of the data instead of searching in the input data space directly. We use generative adversarial networks (GANs) (Goodfellow et al., 2014) to learn a projection to map normally distributed fixed-length vectors to data instances. Given an input instance, we search for adversaries in the neighborhood of its corresponding representation in latent space by sampling within a range that is recursively tightened. Figure 1 provides an example of adversaries for digit recognition. Given a multi-layer perceptron (MLP) for MNIST and an image from test data (Figure 1a), our approach generates a *natural* adversarial example (Figure 1e) which is classified incorrectly as "2" by the classifier. Compared to the adversary generated by the existing Fast Gradient Sign Method (FGSM) (Goodfellow et al., 2015) that adds gradient-based noise (Figures 1c and 1b), our adversary (Figure 1e) looks like a hand-written digit similar to the original input. Further, the difference (Figure 1d) provides some insight into the classifier's behavior, such as the fact that slightly thickening (blue) the bottom stroke and thinning (red) the one above it, fools the classifier.

We apply our approach to both image and text domains, and generate adversaries that are more natural and grammatical, semantically close to the input, and helpful to interpret the local behavior of black-box models. We present examples of natural adversaries for image classification, textual entailment, and machine translation. Experiments and human evaluation also demonstrate that our approach can help evaluate the *robustness* of black-box classifiers, even without labeled training data.

## 2   FRAMEWORK FOR GENERATING NATURAL ADVERSARIES

In this section, we describe the problem setup and details of our framework for generating natural adversarial examples of both continuous images and discrete text data. Given a black-box classifier $f$ and a corpus of unlabeled data $X$, the goal here is to generate adversarial example $x^*$ for a given data instance $x$ that results in a different prediction, i.e. $f(x^*) \neq f(x)$. In general, the instance $x$ may not be in $X$, but comes from the same underlying distribution $\mathcal{P}_x$, which is the distribution we want to generate $x^*$ from as well. We want $x^*$ to be the nearest such instance to $x$ in terms of the manifold that defines the data distribution $\mathcal{P}_x$, instead of in the original data representation.

Unlike other existing approaches that search directly in the input space for adversaries, we propose to search in a corresponding dense representation of $z$ space. In other words, instead of finding the adversarial $x^*$ directly, we find the adversarial $z^*$ in an underlying dense vector space which defines the distribution $\mathcal{P}_x$, and then map it back to $x^*$ with the help of a generative model. By searching for samples in the latent low-dimensional $z$ space and mapping them to $x$ space to identify the adversaries, we encourage these adversaries to be valid (legible for images, and grammatical for sentences) and semantically close to the original input.

**Background: Generative Adversarial Networks** To tackle the problem described above, we need powerful generative models to learn a mapping from the latent low-dimensional representation to the distribution $\mathcal{P}_x$, which we estimate using samples in $X$. GANs are a class of such generative models that can be trained via procedures of minimax game between two competing networks (Goodfellow et al., 2014): given a large amount of unlabeled instances $X$ as training data, the generator $\mathcal{G}_\theta$ learns to map some noise with distribution $p_z(z)$ where $z \in \mathbb{R}^d$ to synthetic data that is as close to the training data as possible; on the other hand, the critic $\mathcal{C}_\omega$ is trained to discriminate the output of the

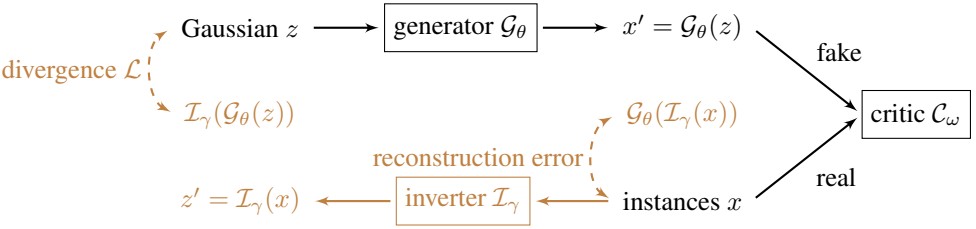

Figure 2: **Training Architecture with a GAN and an Inverter.** Loss of the inverter combines reconstruction error of $x$ with divergence between Gaussian distribution $z$ and $\mathcal{I}_\gamma(\mathcal{G}_\theta(z))$.

generator from real data samples from $X$. The original objective function of GANs has been found to be hard to optimize in practice, for reasons theoretically investigated in Arjovsky & Bottou (2017). Arjovsky et al. (2017) refine the objective with Wasserstein-1 distance as:

$$\min_\theta \max_\omega \mathbb{E}_{x \sim p_x(x)}[\mathcal{C}_\omega(x)] - \mathbb{E}_{z \sim p_z(z)}[\mathcal{C}_\omega(\mathcal{G}_\theta(z))]. \tag{1}$$

Wasserstein GAN achieves improvement in the stability of learning and provides useful learning curves. A number of further improvements to the GAN framework have been introduced (Salimans et al., 2016; Arjovsky & Bottou, 2017; Gulrajani et al., 2017; Rosca et al., 2017) that we discuss in Section 6. We incorporate the structure of WGAN and relevant improvements as a part of our framework for generating natural examples close to the training data distribution, as we describe next.

**Natural Adversaries** In order to represent natural instances of the domain, we first train a WGAN on corpus $X$, which provides a *generator* $\mathcal{G}_\theta$ that maps random dense vectors $z \in \mathbb{R}^d$ to samples $x$ from the domain of $X$. We separately train a matching *inverter* $\mathcal{I}_\gamma$ to map data instances to corresponding dense representations. As in Figure 2, we minimize the reconstruction error of $x$, and the divergence between sampled $z$ and $\mathcal{I}_\gamma(\mathcal{G}_\theta(z))$ to encourage the latent space to be normally distributed:

$$\min_\gamma \mathbb{E}_{x \sim p_x(x)} \|\mathcal{G}_\theta(\mathcal{I}_\gamma(x)) - x\| + \lambda \cdot \mathbb{E}_{z \sim p_z(z)}[\mathcal{L}(z, \mathcal{I}_\gamma(\mathcal{G}_\theta(z)))]. \tag{2}$$

Using these learned functions, we define the *natural adversarial example* $x^*$ as the following:

$$x^* = \mathcal{G}_\theta(z^*) \text{ where } z^* = \arg\min_{\tilde{z}} \|\tilde{z} - \mathcal{I}_\gamma(x)\| \text{ s.t. } f(\mathcal{G}_\theta(\tilde{z})) \neq f(x). \tag{3}$$

Instead of $x$, we perturb its dense representation $z' = \mathcal{I}_\gamma(x)$, and use the generator to test whether a perturbation $\tilde{z}$ fools the classifier by querying $f$ with $\tilde{x} = \mathcal{G}_\theta(\tilde{z})$. Figure 3 shows our generation process. A synthetic example is included for further intuition in Appendix A. As for the divergence $\mathcal{L}$, we use $L_2$ distance with $\lambda = .1$ for images and Jensen-Shannon distance with $\lambda = 1$ for text data.

**Search Algorithms** We propose two approaches to identify the adversary (pseudocode in Appendix B), both of which utilize the inverter to obtain the latent vector $z' = \mathcal{I}_\gamma(x)$ of $x$, and feed perturbations $\tilde{z}$ in the neighborhood of $z'$ to the generator to generate natural samples $\tilde{x} = \mathcal{G}_\theta(\tilde{z})$. In *iterative stochastic search* (Algorithm 1), we incrementally increase the search range (by $\Delta r$) within which the perturbations $\tilde{z}$ are randomly sampled ($N$ samples for each iteration), until we have generated samples $\check{x}$ that change the prediction. Among these samples $\check{x}$, we choose the one which has the closest $z^*$ to the original $z'$ as an adversarial example $x^*$. To improve the efficiency beyond this naive search, we propose a coarse-to-fine strategy we call *hybrid shrinking search* (Algorithm 2). We first search for adversaries in a wide search range, and recursively tighten the upper bound of the search range with denser sampling in bisections. Extra iterative search steps are taken to further

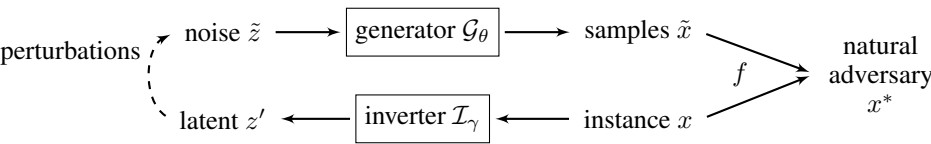

Figure 3: **Natural Adversary Generation.** Given an instance $x$, our framework generates natural adversaries by perturbing inverted $z'$ and decoding perturbations $\tilde{z}$ via $\mathcal{G}_\theta$ to query the classifier $f$.

Table 1: **Adversarial examples of MNIST.** The top row shows images from original test data, and the others show corresponding adversaries generated by FGSM against LeNet and our approach against both RF and LeNet. Predictions from the classifier are shown in the corner of each image.

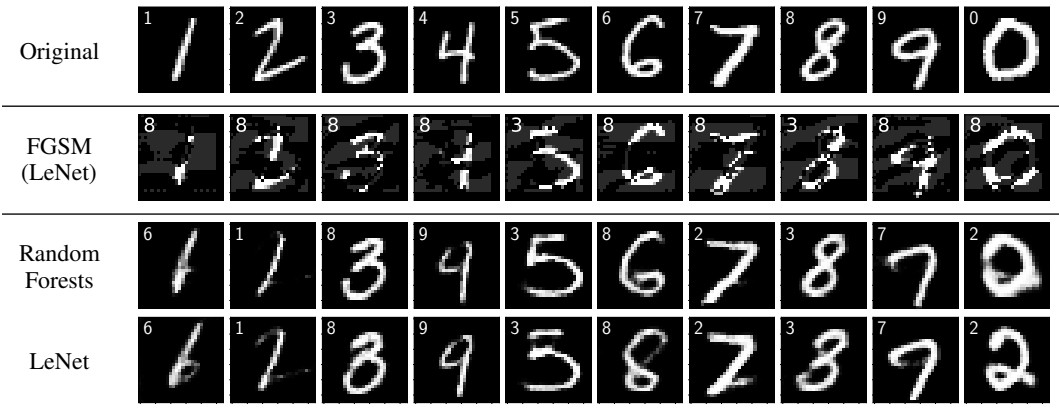

tighten the upper bound of the optimal $\Delta z$. With the hybrid shrinking search in Algorithm 2, we observe a $4\times$ speedup while achieving similar results as Algorithm 1. Both these search algorithms are sample-based and applicable to black-box classifiers with no need of access to their gradients. Further, they are guaranteed to find an adversary, i.e. one that upper bounds the optimal adversary.

## 3 ILLUSTRATIVE EXAMPLES

We demonstrate the potential of our approach (Algorithm 1) in generating informative, legible, and natural adversaries by applying it to a number of classifiers for both visual and textual domains.

### 3.1 GENERATING IMAGE ADVERSARIES

Image classification has been a focus for adversarial example generation due to the recent successes in computer vision. We apply our approach to two standard datasets, MNIST and LSUN, and present generated natural adversaries. We use $\Delta r = 0.01$ and $N = 5000$ with model details in Appendix C.

**Handwritten Digits** Scans of human-written text provide an intuitive definition of what is *natural*, i.e. do the generated images look like something a person would write? In other words, how would a human change a digit in order to fool a classifier? We train a WGAN with $z \in \mathbb{R}^{64}$ on 60,000 MNIST images following similar procedures as in Gulrajani et al. (2017), with the generator consisting of transposed convolutional layers and the critic consisting of convolutional layers. We include the inverter with fully connected layers on top of the critic's last hidden layer. We train two target classifiers to generate adversaries against: Random Forests (RF) with 5 trees (test accuracy 90.45%), and LeNet, as trained in LeCun et al. (1998) (test accuracy 98.71%). We treat both these classifiers as black-boxes, and present the generated adversaries in Table 1 with examples of each digit (from test instances that the GAN or classifiers never observed). Adversaries generated by FGSM look like the original digits eroded by uninterpretable noise (these may not be representative of the approach, as changing $\epsilon$ for the method results in substantially different results). Our *natural* adversaries against both classifiers are quite similar to the original inputs in overall style and shape, yet provide informative insights into classifiers' decision behavior around the input. Take the digit "5" as an example: dimming the vertical stroke can fool LeNet into predicting "3". Further we observe that adversaries against RF often look closer to the original images in overall shape than those against LeNet. Although generating as impressive natural adversaries against more accurate LeNet is difficult, it implies that compared to RF, LeNet requires more substantial changes to the inputs to be fooled; in other words, RF is less robust than LeNet in classification. We will return to this observation later.

**Church vs Tower** We apply our approach to outdoor, color images of higher resolution. We choose the category of "Church Outdoor" in LSUN dataset (Yu et al., 2015), randomly sample the same amount of 126,227 images from the category of "Tower", and resize them to resolution of 64×64.

Table 2: **Adversarial examples against MLP classifier of LSUN by our approach.** 4 original images each of "Church" and "Tower", with their adversaries of the flipped class in the bottom row.

Table 3: **Textual Entailment.** For a pair of premise ($\mathbf{p}$ : ) and hypothesis ($\mathbf{h}$ : ), we present the generated adversaries for three classifiers by perturbing the hypothesis ($\mathbf{h'}$ : ). The last column provides the true label, followed by the changes in the prediction for each classifier.

| Classifiers | Sentences | Label |
|---|---|---|
| Original | $\mathbf{p}$ : The man wearing blue jean shorts is grilling. 
 $\mathbf{h}$ : The man is walking his dog. | Contradiction |
| Embedding | $\mathbf{h'}$ : The man is walking by the dog. | Contradiction $\rightarrow$ Entailment |
| LSTM | $\mathbf{h'}$ : The person is walking a dog. | Contradiction $\rightarrow$ Entailment |
| TreeLSTM | $\mathbf{h'}$ : A man is winning a race. | Contradiction $\rightarrow$ Neutral |

The training procedure is similar to MNIST, except that the generator and critic in WGAN are deep residual networks (He et al., 2016) and $z \in \mathbb{R}^{128}$. We train an MLP classifier on these two classes with test accuracy of 71.3%. Table 2 presents original images for both classes and corresponding adversarial examples. From looking at these pairs, we can observe that the generated adversaries make changes that are natural for this domain. For example, to change the classifier's prediction from "Church" to "Tower", the adversaries sharpen the roof, narrow the buildings, or change a tree into a tower. We can observe similar behavior in the other direction: the image with the Eiffel Tower is changed to a "church" by converting a woman into a building, and narrowing the tower.

## 3.2 GENERATING TEXT ADVERSARIES

Generating grammatical and linguistically coherent adversarial sentences is a challenging task due to the discrete nature of text: adding *imperceptible* noise is impossible, and most actual changes to $x$ may not result in grammatical text. Prior approaches on generating textual adversaries (Li et al., 2016; Alvarez-Melis & Jaakkola, 2017; Jia & Liang, 2017) perform word erasures and replacements directly on text input space $x$, using domain-specific rule based or heuristic based approaches, or require manual intervention. Our approach, on the other hand, performs perturbations in the continuous space $z$, that has been trained to produce semantically and syntactically coherent sentences automatically.

We use the adversarially regularized autoencoder (ARAE) (Zhao et al., 2017) for encoding discrete text into continuous codes. ARAE model encodes a sentence with an LSTM encoder into continuous code and then performs adversarial training on these codes to capture the data distribution. We introduce an inverter that maps these continuous codes into the Gaussian space of $z \in \mathbb{R}^{100}$. We use a 4-layer strided CNN for the encoder as it yields more coherent sentences than LSTMs from the ARAE model, however LSTM works well as the decoder. We train two MLP models for the generator and the inverter, to learn mappings between noise and continuous codes. We train our framework on the Stanford Natural Language Inference (SNLI) (Bowman et al., 2015) data of 570k labeled human-written English sentence pairs with the same preprocessing as Zhao et al. (2017), using $\Delta r =$ 0.01 and $N = 100$. We present details of the architecture and sample perturbations in Appendix D.

**Textual Entailment** Textual Entailment (TE) is a task designed to evaluate common-sense reasoning for language, requiring both natural language understanding and logical inferences for text snippets. In this task, we classify a pair of sentences, a *premise* and a *hypothesis*, into three categories depending on whether the hypothesis is *entailed* by the premise, *contradicts* the premise, or is *neutral* to it. For

Table 4: **Machine Translation.** "Adversary" that introduces the word "stehen" into the translation.

| Source Sentence (English) | Generated Translation (German) |
|---|---|
| **s** : A man and woman **sitting** on the sidewalk. | Ein Mann und eine Frau, die auf dem Bürgersteig **sitzen**. |
| **s**′ : A man and woman **stand** on the bench. | Ein Mann und eine Frau **stehen** auf der Bank. |

Table 5: **"Adversaries" to find dropped verbs.** The left column contains the original sentence $s$ and its adversary $s'$, while the right contains their translations, with English translation in red.

| Source Sentence (English) | Generated Translation (German) |
|---|---|
| **s** : People sitting in a dim restaurant **eating**. | Leute, die in einem dim Restaurant **essen** sitzen. |
| **s**′ : People sitting in a living room **eating**. | Leute, die in einem Wohnzimmeressen sitzen. 
 *(People sitting in a living room.)* |
| **s** : Elderly people **walking** down a city street. | Ältere Menschen, die eine Stadtstraße **hinuntergehen**. |
| **s**′ : A man **walking** down a street playing. | Ein Mann, der eine Straße entlang spielt. 
 *(A man playing along a street.)* |

instance, the sentence "There are children present" is entailed by the sentence "Children smiling and waving at camera", while the sentence "The kids are frowning" contradicts it. We use our approach to generate adversaries by perturbing the hypothesis to deceive classifiers, keeping the premise unchanged. We train three classifiers of varying complexity, namely, an *embedding* classifier that is a single layer on top of the average word embeddings, an *LSTM* based model consisting of a single layer on top of the sentence representations, and *TreeLSTM* (Chen et al., 2017) that uses a hierarchical LSTM on the parses and is a top-performing classifier for this task. A few examples comparing the three classifiers are shown in Table 3 (more examples in Appendix D.1). Although all classifiers correctly predict the label, as the classifiers get more accurate (from *embedding* to *LSTM* to *TreeLSTM*), they require much more substantial changes to the sentences to be fooled.

**Machine translation** We consider machine translation not only because it is one of the most successful applications of neural approaches to NLP, but also since most practical translation systems lie behind black-box access APIs. The notion of *adversary*, however, is not so clear here as the output of a translation system is not a class. Instead, we define adversary for machine translation relative to a *probing function* that tests the translation for certain properties, ones that may lead to linguistic insights into the languages, or detect potential vulnerabilities. We use the same generator and inverter as in entailment, and find such "adversaries" via API access to the currently deployed Google Translate model (as of *October 15, 2017*) from English to German.

First, let us consider the scenario in which we want to generate adversarial English sentences such that a specific German word is introduced into the German translation. The probing function here would test the translation for the presence of that word, and we would have found an adversary (an English sentence) if the probing function *passes* for a translation. We provide an example of such a probing function that introduces the word "stehen" ("stand" in English) to the translation in Table 4 (more examples in Appendix D.2). Since the translation system is quite strong, such adversaries are not surfacing the vulnerabilities of the model, but instead can be used as a tool to understand or learn different languages (in this example, help a German speaker learn English).

We can design more complex probing functions as well, especially ones that target specific vulnerabilities of the translation system. Let us consider translations of English sentences that contain two active verbs, e.g. "People sitting in a restaurant eating", and see that the German translation has the two verbs as well, "essen" and "sitzen", respectively. We now define a probing function that passes only if the perturbed English sentence $s'$ contains both the verbs, but the translation only has one of them. An adversary for such a probing function will be an English sentence ($s'$) that is similar to the original sentence ($s$), but for some reason, its translation is missing one of the verbs. Table 5 presents examples of generated adversaries using such a probing function (with more in Appendix D.2). For example, one that tests whether "essen" is dropped from the translation when its English counterpart "eating" appears in the source sentence ("People sitting in a living room eating."). These adversaries thus suggest a vulnerability in Google's English to German translation system: a word acting as a gerund in English often gets dropped from the translation.

Table 6: **Statistics of adversaries against models for both MNIST and TE.** We include the average $\Delta z$ for the adversaries and the proportion where each classifier's adversary has the largest $\Delta z$ compared to the others for the same instance (significant with $p < 0.0005$ using the sign test). The higher values correspond to stronger robustness, as is demonstrated by higher test accuracy.

|  |  | **Average $\Delta z$** | **P(largest $\Delta z$)** | **Test accuracy (%)** |
|---|---|---|---|---|
| **MNIST** | Random Forests | 1.24 | 0.22 | 90.45 |
|  | LeNet | 1.61 | 0.78 | 98.71 |
| **Entailment** | Embeddings | 0.12 | 0.15 | 62.04 |
|  | LSTM | 0.14 | 0.18 | 69.60 |
|  | TreeLSTM | 0.26 | 0.66 | 89.04 |

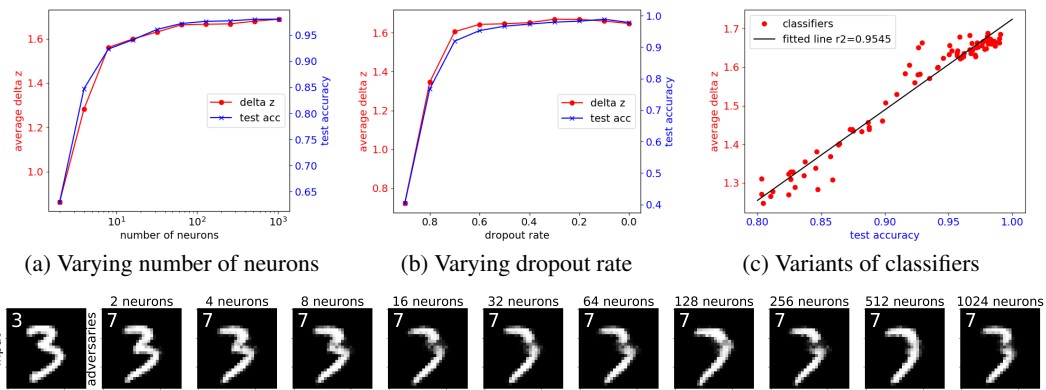

(a) Varying number of neurons    (b) Varying dropout rate    (c) Variants of classifiers

(d) Adversaries for the same input digit against 10 classifiers with increasing capacity.

Figure 4: **Classifier accuracy and average $\Delta z$ of their adversaries.** In (a) and (b) we vary the number of neurons and dropout rate, respectively. In (c) we present the correlation between accuracy and average $\Delta z$ for 80 different classifiers. (d) shows adversaries for an input image, against a set of classifiers with a single hidden layer, but varying number of neurons.

## 4 EXPERIMENTS

In this section, we demonstrate that our approach can be utilized to compare and evaluate the *robustness* of black-box models even without labeled data. We present experimental results on images and text data with evaluations from both statistical analysis and pilot user studies.

**Robustness of Black-box Classifiers** We apply our framework to various black-box classifiers for both images and text, and observe that it is useful for evaluating and interpreting these models via comparisons. The primary intuition behind this analysis is that more accurate classifiers often require more substantial changes to the instance to change their predictions, as noted in the previous section. In the following experiments, we apply the more efficient *hybrid shrinking search* (Algorithm 2).

In order to quantify the extent of change for an adversary, the change in the original $x$ representation may not be meaningful, such as RMSE of the pixels or string edit distances, for the same reason we are generating *natural* adversaries: they do not correspond to the *semantic* distance underlying the data manifold. Instead we use the distance of the adversary in the latent space, i.e. $\Delta z = \|z^* - z'\|$, in order to measure how much each adversary is modified to change the classifier prediction. We also consider the set of adversaries generated for each instance against a group of classifiers, and count how many times the adversary of each classifier has the highest $\Delta z$. We present these statistics in Table 6 for both MNIST (over 100 test images, 10 per digit) and Textual Entailment (over 1260 test sentences), against the classifiers we described in Section 3. For both the tasks, we observe that more accurate classifiers require larger changes to the inputs (by both measures), indicating that generating such adversaries, even for unlabeled data, can evaluate the accuracy of black-box classifiers.

| Table 7: **Pilot study with MNIST** | RF | LeNet |
|---|---|---|
| Looks handwritten? | 0.88 | 0.71 |
| Which closer to original? | 0.87 | 0.13 |

| Table 8: **Pilot study with Textual Entailment** | LSTM | TreeLSTM |
|---|---|---|
| Is adversary grammatical? | 0.86 | 0.78 |
| Is it similar to the original? | 0.81 | 0.58 |

We now consider evaluation on a broader set of classifiers, and study the effect of changing hyper-parameters of models on the results (focusing on MNIST). We train a set of neural networks with one hidden layer by varying the number of neurons exponentially from 2 to 1024. In Figure 4a, we observe that the average $\Delta z$ of adversaries against these models has a similar trend as their test accuracy. The generated adversaries for a single digit "3" in Figure 4d verify this observation: the adversaries become increasingly different from the original input as classifiers become more complex. We provide similar analysis by fixing the model structure but varying the dropout rates from 0.9 to 0.0 in Figure 4b, and observe a similar trend. To confirm that this correlation holds generally, we train 80 total classifiers that differ in the layer sizes, regularization, and amount of training data, and plot their test set accuracy against the average magnitude of change in their adversaries in Figure 4c. Given this strong correlation, we are confident that our framework for generating natural adversaries can be useful for automatically evaluating black-box classifiers, even in the absence of labeled data.

**Human Evaluation**  We carry out a pilot study with human subjects to evaluate how natural the generated adversaries are, and whether the adversaries they think are similar to the original ones correspond with the less accurate classifiers (as in the evaluation above). For both image classification and textual entailment, we select a number of instances randomly, generate adversaries for each against two classifiers, and present a questionnaire to the subjects that evaluates: (1) how natural or legible each generated adversary is; (2) which of the two adversaries is closer to the original instance.

For hand-written digits from MNIST, we pick 20 images (2 for each digit), generate adversaries against RF and LeNet (two adversaries for each image), and obtain 13 responses for each of the questions. In Table 7, we see that the subjects agree that our generated adversaries are quite natural, and also, they find RF adversaries to be much closer to the original image than LeNet (i.e. more accurate classifiers, as per test accuracy on their provided labels, have more distant adversaries). We also compare adversaries against LeNet generated by FGSM and our approach, and find that 78% of the time the subjects agree that our adversaries make changes to the original images that are more natural (it is worth noting that FGSM is not applicable to RF for comparison). We carry out a similar pilot study for the textual entailment task to evaluate the quality of the perturbed sentences. We present a set of 20 pairs of sentences (premise and hypothesis), and adversarial hypotheses against both LSTM and TreeLSTM classifiers, and receive 4 responses for each of the questions above. The results in Table 8 also validate our previous results: the generated sentences are found to be grammatical and legible, and classifiers that need more substantial changes to the hypothesis tend to be more accurate. We leave a more detailed user study for future work.

## 5  RELATED WORK

The fast gradient sign method (FGSM) has been proposed in Goodfellow et al. (2015) to generate adversarial examples *fast* rather than optimally. Intuitively, the method shifts the input by $\epsilon$ in the direction of minimizing the cost function. Kurakin et al. (2016) propose a simple extension of FGSM by applying it multiple times, which generates adversarial examples with a higher attack rate, but the underlying idea is the same. Another method known as the Jacobian-based saliency map attack (JSMA) has been introduced by Papernot et al. (2016b). Unlike FGSM, JSMA generates adversaries by greedily modifying the input instance feature-wise. A saliency map is computed with gradients to indicate how important each feature is for the prediction, and the most important one is modified repeatedly until the instance changes the resulting classification. Moreover, it has been observed in practice that adversarial examples designed against a model are often likely to successfully attack another model for the same task that has not been given access to. This transferability property of adversarial examples makes it more practical to attack and evaluate deployed machine learning systems in realistic scenarios (Papernot et al., 2016a; 2017).

All these attacks above are based on gradients with access to the parameters of differentiable classifiers. Moosavi-Dezfooli et al. (2017) try to find a single noise vector which can cause imperceptible changes

in most of data points, and meanwhile reduce the classifier accuracy significantly. Our method is capable of generating adversaries against black-box classifiers, even those without gradients such as Random Forests. Also, the noise added by these methods is uninterpretable, while the natural adversaries generated by our approach provide informative insights into classifiers' decision behavior.

Due to the discrete domains involved in text, adversaries for text have received less attention. Jia & Liang (2017) generate adversarial examples for evaluating reading comprehension systems with predefined rules and candidate words for substitution after analyzing and rephrasing the input sentences. Li et al. (2016) introduce a framework to understand neural network through different levels of representation erasure. However, erasure of words or phrases directly often harms text integrity, resulting in semantically or grammatically incorrect sentences. Ribeiro et al. (2018) replace tokens by random words of the same POS tag with probability proportional to embedding similarity. Belinkov & Bisk (2018) explore approaches to increase the robustness of character-based machine translation models on text corrupted with character-level noise. With the help of expressive generative models, our approach instead perturbs the latent coding of sentences, resulting in legible generated sentences that are grammatical and semantically similar to the original input. These merits make our framework suitable for text applications such as sentiment analysis, textual entailment, and machine translation.

## 6 DISCUSSION AND FUTURE WORK

Our framework builds upon GANs as the generative models, and thus the capabilities of GANs directly effects the quality of generated examples. In visual domains, although there have been lots of appealing results produced by GANs, the training is well known to be brittle. Many recent approaches address how to improve the training stability and the objective function of GANs (Salimans et al., 2016; Arjovsky et al., 2017). Gulrajani et al. (2017) further improve the training of WGAN with regularization of gradient penalty instead of weight clipping. In our practice, we observe that we need to carefully balance the capacities of the generator, the critic, and the inverter that we introduced, to avoid situations such as model collapse. For natural languages, because of the discrete nature and non-differentiability, applications related to text generation have been relatively less studied. Zhao et al. (2017) propose to incorporate a discrete structure autoencoder with continuous code space regularized by WGAN for text generation. Given that there are some concerns about whether GANs actually learn the distribution (Arora & Zhang, 2017), it is worth noting that we can also incorporate other generative models such as Variational Auto-Encoders (VAEs) (Kingma & Welling, 2014) into our framework, as used in Hu et al. (2017) to generate text with controllable attributes, which we will explore in the future. We focus on GANs because adversarial training often results in higher quality images, while VAEs tend to produce blurrier ones (Goodfellow, 2016). We also plan to apply the fusion and variant of VAEs and GANs such as $\alpha$-GAN in Rosca et al. (2017) and Wasserstein Auto-Encoders in Tolstikhin et al. (2018). Note that as more advanced GANs are introduced to address these issues, they can be directly incorporated into our framework.

Our *iterative stochastic search* algorithm for identifying adversaries is computationally expensive since it is based on naive sampling and local-search. Search based on gradients such as FGSM are not applicable to our setup because of black-box classifiers and discrete domain applications. We improve the efficiency with *hybrid shrinking search* by using a coarse-to-fine strategy that finds the upper-bounds by using fewer samples, and then performs finer search in the restricted range. We observe around $4\times$ speedup with this search while achieving similar results as the iterative search. The accuracy of our inverter mapping the input to its corresponding dense vector in latent space is also important for searching adversaries in the right neighborhood. In our experiments, we find that fine-tuning the latent vector produced by the inverter with a fixed GAN can further refine the generated adversarial examples, and we will investigate other such extensions of the search in future. There is an implicit assumption in this work that the generated samples are within the same class if the added perturbations are small enough, and the generated samples look as if they belong to different classes when the perturbations are large. However, note that it is also the case for FGSM and other such approaches: when their $\epsilon$ is small, the noise is imperceptible; but with a large $\epsilon$, one often finds noisy instances that might be in a different class (see Table 1, digit 8 for an example). While we do observe this behavior in some cases, the corresponding classifiers require much more substantial changes to the input, which is why we can utilize our approach to evaluate black-box classifiers.

## 7 CONCLUSIONS

In this paper, we propose a framework for generating *natural* adversaries against black-box classifiers, and apply the same approach to both visual and textual domains. We obtain adversaries that are legible, grammatical, and meaningfully similar to the input. We show that these natural adversaries can help in interpreting the decision behavior and evaluating the accuracy of black-box classifiers even in absence of labeled training data. We use our approach, built upon recent work in GANs, to generate adversaries for a wide range of applications including image classification, textual entailment, and machine translation (via the Google Translate API). Code used to generate such natural adversaries is available at `https://github.com/zhengliz/natural-adversary`.

### ACKNOWLEDGMENTS

We would like to thank Ananya, Casey Graff, Eric Nalisnick, Pouya Pezeshkpour, Robert Logan, and the anonymous reviewers for the discussions and feedback on earlier versions. We would also like to thank Ishaan Gulrajani and Junbo Jake Zhao for making their code available. This work is supported in part by Adobe Research and in part by FICO. The views expressed are those of the authors and do not reflect the official policy or position of the funding agencies.

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

APPENDIX

## A    ILLUSTRATION WITH SYNTHETIC DATA

As shown in Figure 5 with a toy example of synthetic data, we can effectively map data instance $x$ to its corresponding latent dense vector $z'$ with the help of the inverter via $\mathcal{I}_\gamma(x)$, and then reconstruct $x$ with the help of the generator via $\mathcal{G}_\theta(\mathcal{I}_\gamma(x))$. For a naive classifier with the horizontal line as decision boundary, adversarial examples should be points above the line given input data $x$ in Figure 5c. By searching in corresponding latent space, our approach finds $x^*$ on the left as a *natural* adversary because it is the closest one in semantic space (along the curve trace) and it exists within the data manifold. However, the gradient-based approaches may find the $x^*$ right above $x$ as adversarial in input space regardless of the actual data distribution.

## B    ALGORITHMS

Algorithm 1 shows the pseudocode of the iterative search of our framework. Starting from the corresponding $z$ of the input instance $x$, we iteratively move the search range outward in latent space until we have generated samples that change the prediction of the classifier $f$. We improve the efficiency with Algorithm 2 by using a coarse-to-fine strategy and combining recursive and iterative search. We first search for adversaries in a wide search range, and recursively tighten the upper bound of the search range with denser sampling in bisections. Extra iterative search steps are taken to further tighten the upper bound of the optimal $\Delta z$. This hybrid shrinking search approach, shown in detail in Algorithm 2, is four times faster to achieve similar adversaries as the iterative search.

## C    ARCHITECTURE FOR CONTINUOUS IMAGES

Figure 3 shows the architecture of our framework for continuous images. We adopt WGAN (Arjovsky et al., 2017) with the objective function in Equation 1, and apply gradient penalty as proposed in Gulrajani et al. (2017). On top of the generator obtained from WGAN, we train an inverter by optimizing Equation 2. For handwritten digits from MNIST dataset, we train a WGAN of latent $z \in \mathbb{R}^{64}$, with a generator consisting of 3 transposed convolutional layers and ReLU activation, and a critic consisting of 3 convolutional layers with filter sizes (64, 128, 256) and strides (2, 2, 2). We include an inverter with 2 fully connected layers of dimensions (4096, 1024) on top of the critic's last hidden layer. For "Church Outdoor" and "Tower" images from LSUN dataset, we follow similar procedures as in Gulrajani et al. (2017) training a WGAN of latent $z \in \mathbb{R}^{128}$. The generator and critic are both residual networks. We use pre-activation residual blocks with two $3 \times 3$ convolutional layers each and ReLU activation. The critic of 4 residual blocks performs downsampling using mean pooling after the second convolution, while the generator contains 4 residual blocks performing nearest-neighbor upsampling before the second convolution. We include an inverter with 3 fully connected layers of dimensions (8192, 2048, 512) on top of the critic's last hidden layer.

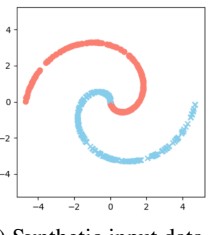
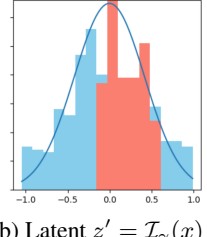
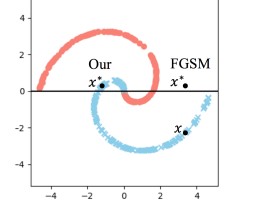

| (a) Synthetic input data $x$ | (b) Latent $z' = \mathcal{I}_\gamma(x)$ | (c) Reconstructed data $\mathcal{G}_\theta(\mathcal{I}_\gamma(x))$ |

Figure 5: **Illustration with synthetic data.** With training data that lies on a complex manifold (a), the inverter maps input to compact *gaussian* latent $z' = \mathcal{I}_\gamma(x)$ in (b), while the generator reconstructs the data via $\mathcal{G}_\theta(\mathcal{I}_\gamma(x))$ in (c). Given $f$ as a binary classifier with decision boundary as the horizontal line in (c), for an input $x$, our approach returns $x^*$ on the left as *natural* adversary that lies on the manifold, while existing approaches may find $x^*$ on the right as the adversary, which is the nearest but impossible.

---

**Algorithm 1** Iterative stochastic search in latent space for adversaries

---

**Require:** a target black-box classifier $f$, an input instance $x$, and a corpus of relevant data $X$
1: **Hyper-parameters:** $N$: number of samples in each iteration, $\Delta r$: increment of search range
2: Train a generator $\mathcal{G}_\theta$ and an inverter $\mathcal{I}_\gamma$ on $X$
3: $y \leftarrow f(x)$, $z' \leftarrow \mathcal{I}_\gamma(x)$, radius $r \leftarrow 0$
4: **loop**                                                                    ▷ loop till we find an adversary
5:     $S \leftarrow \emptyset$
6:     **for** sample $N$ random noise vectors $\epsilon$ of norms within $(r, r + \Delta r]$ **do**
7:         $\tilde{z} \leftarrow z' + \epsilon$, $\tilde{x} \leftarrow \mathcal{G}_\theta(\tilde{z})$, $\tilde{y} \leftarrow f(\tilde{x})$        ▷ perturbation, sample generation, prediction
8:         **if** $\tilde{y} \neq y$ **then**
9:             $S \leftarrow S \cup \langle \tilde{x}, \tilde{y}, \tilde{z} \rangle$
10:    **if** $S = \emptyset$ **then**                                          ▷ no adversary generated
11:        $r \leftarrow r + \Delta r$                                          ▷ move search range outward
12:    **else**                                                                ▷ certain adversary generated
13:        **return** $\langle x^*, y^*, z^* \rangle = \text{argmin}_{\langle \tilde{x}, \tilde{y}, \tilde{z} \rangle \in S} \|\check{z} - z'\|$        ▷ return the closest sample

---

**Algorithm 2** Hybrid shrinking search in latent space for adversaries

---

**Require:** a target black-box classifier $f$, an input instance $x$, and a corpus of relevant data $X$
1: **Hyper-parameters:** $N$: number of samples in each iteration, $\Delta r$: increment of search range, $B$: limit of iterations, $r$: upper limit of search range
2: Train a generator $\mathcal{G}_\theta$ and an inverter $\mathcal{I}_\gamma$ on $X$
3: $y \leftarrow f(x)$, $z' \leftarrow \mathcal{I}_\gamma(x)$, $l \leftarrow 0, i \leftarrow 0$
4: **First, recursive search:**
5: **while** $r - l \geq \Delta r$ **do**
6:     $S \leftarrow \emptyset$
7:     **for** sample $N$ random noise vectors $\epsilon$ of magnitude within $(l, r]$ **do**
8:         $\tilde{z} \leftarrow z' + \epsilon$, $\tilde{x} \leftarrow \mathcal{G}_\theta(\tilde{z})$, $\tilde{y} \leftarrow f(\tilde{x})$
9:         **if** $\tilde{y} \neq y$ **then**
10:            $S \leftarrow S \cup \langle \tilde{x}, \tilde{y}, \tilde{z} \rangle$
11:    **if** $S = \emptyset$ **then**                                          ▷ no adversary generated
12:        $l \leftarrow (l + r)/2$                                            ▷ shrink search range by half
13:    **else**                                                                ▷ certain adversary generated
14:        $\langle x^*, y^*, z^* \rangle = \text{argmin}_{\langle \tilde{x}, \tilde{y}, \tilde{z} \rangle \in S} \|\check{z} - z'\|$        ▷ store the closest sample
15:        $l \leftarrow 0, r \leftarrow \|z^* - z'\|$                          ▷ update upper bound of $\Delta z$
16: **Then, iterative search:**
17: **while** $i < B$ and $r > 0$ **do**
18:    $S \leftarrow \emptyset, l \leftarrow \max(0, r - \Delta r)$
19:    **for** sample $N$ random noise vectors $\epsilon$ of norms within $(l, r]$ **do**
20:        $\tilde{z} \leftarrow z' + \epsilon$, $\tilde{x} \leftarrow \mathcal{G}_\theta(\tilde{z})$, $\tilde{y} \leftarrow f(\tilde{x})$
21:        **if** $\tilde{y} \neq y$ **then**
22:            $S \leftarrow S \cup \langle \tilde{x}, \tilde{y}, \tilde{z} \rangle$
23:    **if** $S = \emptyset$ **then**
24:        $i \leftarrow i + 1, r \leftarrow r - \Delta r$                      ▷ increase counter, continue searching
25:    **else**
26:        $\langle x^*, y^*, z^* \rangle = \text{argmin}_{\langle \tilde{x}, \tilde{y}, \tilde{z} \rangle \in S} \|\check{z} - z'\|$        ▷ store the closest sample
27:        $i \leftarrow 0, r \leftarrow \|z^* - z'\|$                          ▷ reset counter, update upper bound of $\Delta z$
28: **return** $\langle x^*, y^*, z^* \rangle$

---

# D ARCHITECTURE FOR DISCRETE TEXT

We use the adversarially regularized autoencoder (ARAE) (Zhao et al., 2017) for encoding discrete text into continuous codes as shown in Figure 6. ARAE model encodes a sentence with an LSTM encoder into continuous code and performs adversarial training on the codes generated from noise and data to approximate the data distribution. We introduce an inverter that maps these continuous codes

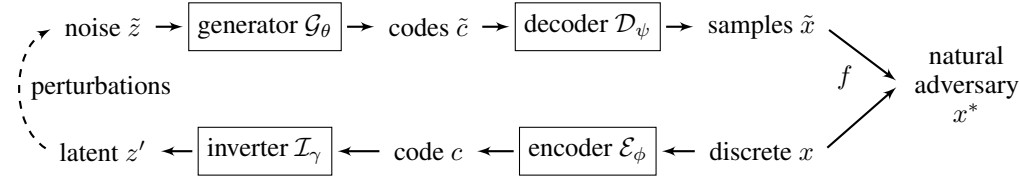

Figure 6: **Model Architecture for Text.** Our model incorporates in the adversarially regularized autoencoder (ARAE) (Zhao et al., 2017) for encoding discrete $x$ into continuous code $c$ and decoding continuous $\tilde{c}$ into discrete $\tilde{x}$ when generating samples.

into the Gaussian space of $z \in \mathbb{R}^{100}$. We use 4 layers of CNN with varying filter sizes (300, 500, 700, and 1000), strides (2, 2, 2) and context windows (5, 5, 3) for encoding text $x$, into continuous space $c \in \mathbb{R}^{300}$. For the decoder, we use a single-layer LSTM with hidden dimension of 300. We also train two MLPs, one each for the generator and the inverter, to learn mappings from noise to continuous codes and continuous codes to noise respectively. The loss functions for different components of the ARAE model, which are autoencoder reconstruction loss and WGAN loss functions for generator and critic, are described in Equations (4), (5), (6) respectively. We first train the ARAE components of encoder, decoder and generator using WGAN strategy, followed by the inverter on top of these with loss function in (7), by minimizing the Jensen-Shannon divergence between the inverted continuous codes and noise samples.

$$\min_{\phi,\psi} \mathcal{L}_{\mathcal{E},\mathcal{D}}(\phi,\psi) = \max_{\phi,\psi} \mathbb{E}_x[\log p_\psi(x|\mathcal{E}_\phi(x))] \tag{4}$$

$$\min_{\omega} \mathcal{L}_{\mathcal{C}}(\omega) = \max_{\omega} \mathbb{E}_x[\mathcal{C}_\omega(\mathcal{E}_\phi(x))] - \mathbb{E}_z[\mathcal{C}_\omega(\mathcal{G}_\theta(z))] \tag{5}$$

$$\min_{\phi,\theta} \mathcal{L}_{\mathcal{E},\mathcal{G}}(\phi,\theta) = \min_{\phi,\theta} \mathbb{E}_x[\mathcal{C}_\omega(\mathcal{E}_\phi(x))] - \mathbb{E}_z[\mathcal{C}_\omega(\mathcal{G}_\theta(z))] \tag{6}$$

$$\min_{\gamma} \mathcal{L}_{\mathcal{I}}(\gamma) = \min_{\gamma} \mathbb{E}_x\|\mathcal{G}_\theta(\mathcal{I}_\gamma(\mathcal{E}_\phi(x))) - \mathcal{E}_\phi(x)\| + \mathbb{E}_z[\text{JSD}(z,\mathcal{I}_\gamma(\mathcal{G}_\theta(z)))] \tag{7}$$

We train our framework on the sentences up to length 10 from Stanford Natural Language Inference (SNLI) (Bowman et al., 2015) dataset, with hyper-parameters of $\Delta r = 0.01$ and $N = 100$. Table 9 shows some examples of the perturbations generated automatically by our approach, which are grammatical and semantically close to the original sentences.

### D.1 TEXTUAL ENTAILMENT EXAMPLES

We provide additional examples of generated adversarial hypotheses for sentences from the SNLI corpus in Table 10, which corresponds to the examples in the main text in Table 3.

### D.2 MACHINE TRANSLATION EXAMPLES

We provide additional examples of the two probing functions in Table 11 and Table 12, corresponding to Table 4 and Table 5 in the main text, respectively.

Table 9: **Text perturbations.** Examples are generated by perturbing the origins in semantic space.

| Original | Some dogs are running on a deserted beach. | A man playing an electric guitar on stage. |
|---|---|---|
| Perturbation | Some dogs are running on a grassy field.
Some dogs are walking along a path.
Some dogs are running down a hill.
A dog is running on a grassy field.
A dog is running down a trail. | A man is playing an electric guitar.
A man is playing an acoustic guitar.
A man is playing an accordion.
A man is playing with an electronic device.
A man is playing with an elephant. |

Table 10: **Textual Entailment.** For a pair of premise ($\mathbf{p}$ : ) and hypothesis ($\mathbf{h}$ : ), we present the generated adversaries for three classifiers by perturbing the hypothesis ($\mathbf{h}'$ : ). The last column provides the true label, followed by the changes in the prediction from each classifier.

| Classifiers | Sentences | Label |
|---|---|---|
| Original | $\mathbf{p}$ : The man walks among the large trees.
$\mathbf{h}$ : The man is lost in the woods. | Neutral |
| Embedding | $\mathbf{h}'$ : The man is lost at the woods. | Contradiction $\rightarrow$ Neutral |
| LSTM | $\mathbf{h}'$ : The man is crying in the woods. | Neutral $\rightarrow$ Contradiction |
| TreeLSTM | $\mathbf{h}'$ : The man is lost in a bed. | Neutral $\rightarrow$ Contradiction |

Table 11: **Machine Translation.** "Adversaries" that introduce the word "stehen" into the Google translation system by perturbing English sentences.

| Source Sentence (English) | Generated Translation (German) |
|---|---|
| $\mathbf{s}$ : Asian women are **sitting** in a Restraunt.
$\mathbf{s}'$ : Asian kids are **standing** in a Restraunt. | Asiatische Frauen **sitzen** in einem Restaurant.
Asiatische Kinder **stehen** in einem Restaurant. |
| $\mathbf{s}$ : People **sitting** on the floor.
$\mathbf{s}'$ : People **standing** on the field. | Leute **sitzen** auf dem Boden.
Leute, die auf dem Feld **stehen**. |

Table 12: **"Adversaries" that find dropped verbs in English-To-German translation.** The left column contains the original sentence $s$ and its adversary $s'$. The right column contains the translations of $s$ and $s'$, with English translation provided for legibility.

| Source Sentence (English) | Generated Translation (German) |
|---|---|
| $\mathbf{s}$ : A man looks back while laughing and **walking**.
$\mathbf{s}'$ : A man is laughing **walking** down the ground. | Ein Mann schaut beim Lachen und **Gehen** zurck.
Ein Mann lacht auf dem Boden.
*(A man laughs on the floor.)* |
| $\mathbf{s}$ : She is cooking food while wearing a **dress**.
$\mathbf{s}'$ : She is cooking **dressed** for a wedding. | Sie kocht Essen, whrend sie ein **Kleid** trgt.
Sie kocht fr eine Hochzeit.
*(She cooks for a wedding.)* |

