# OpenReview forum: "Generating Natural Adversarial Examples"
_ICLR.cc/2018/Conference — Accept (Poster)_

### Official Review · AnonReviewer2 · 2017-11-26
**A novel idea for generating more useful adversary examples.**

**Rating:** 7
**Confidence:** 4

**Review:**


Summary:
 A method for creation of semantical adversary examples in suggested. The ‘semantic’ property is measured by building a latent space with mapping from this space to the observable (generator) and back (inverter). The generator is trained with a WGAN optimization. Semantic adversarials examples are them searched for by inverting an example to its sematic encoding and running local search around it in that space. The method is tested for generation of images on MNist and part of LSUM data and for creation of text examples which are adversarial in some sense to inference and translation sentences. It is shown that the distance between adversarial example and the original example in the latent space is proportional to the accuracy of the classifier inspected.
Page 3: It seems that the search algorithm has a additional parameter: r_0, the size of the area in which search is initiated. This should be explicitly said and the parameter value should be stated.
Page 4:
-	the implementation details of the generator, critic and invertor networks are not given in enough details, and instead the reader is referred to other papers. This makes this paper non-clear as a stand alone document, and is a problem for a paper which is mostly based on experiments and their results: the main networks used are not described.
-	the visual examples are interesting, but it seems that they are able to find good natural adversary examples only for a weak classifier. In the MNist case, the examples for thr random forest are nautral and surprising, but those for the LE-Net are often not: they often look as if they indeed belong to the other class (the one pointed by the classifier). In the churce-vs. tower case, a  relatively weak MLP classifier was used. It would be more instructive to see the results for a better, convolutional classifier.
Page 5:
-	the description of the various networks used for text generation is insufficient for understanding:
o	The AREA is described in two sentences. It is not clear how this module is built, was loss was it used to optimize in the first place, and what elements of it are re0used for the current task
o	 ‘inverter’ here is used in a sense which is different than in previous sections of the paper: earlier it denoted the mapping from output (images) to the underlying latent space. Here it denote  a mapping between two latent spaces.
o	 It is not clear what the ‘four-layers strided CNN’ is: its structure, its role in the system. How is it optimized?
o	In general: a block diagram showing the relation between all the system’s components may be useful, plus the details about the structure and optimization of the various modules. It seems that the system here contains 5 modules instead of the three used before (critic, generator and inverter), but this is not clear enough. Also which modules are pre-trained, which are optimized together,a nd which are optimized separately is not clear.
o	SNLI data should be described: content, size, the task it is used for


Pro:
-	A novel idea of producing natural adversary examples with a GAN
-	The generated examples are in some cases useful for interpretation and network understanding
-	The method enables creation of adversarial examples for block box classifiers
Cons
-	The idea implementation is basic. Specifically search algorithm presented is quite simplistic, and no variations other than plain local search were developed and tested
-	The generated adversarial examples created for successful complex classifiers are often not impressive and useful (they are either not semantical, or semantical but correctly classified by the classifier). Hence It is not clear if the latent space used by the method enables finding of interesting adversarial examples for accurate classifiers.

---

> ### Author Response · Authors · 2017-12-15
> **Response to Reviewer 2**
>
> Thanks for the review.
>
> Details: We held out a lot of implementation details due to the space constraints, but will gladly incorporate them in subsequent versions. We will include some of the more important ones you mentioned in the next revision, with the rest in the appendix. In the first step of the search algorithm, it samples from the range of (0, \Delta r] with r_0 = 0. The Stanford Natural Language Inference (SNLI) corpus is a collection of 570k human-written English sentence pairs manually labeled with whether each hypothesis is entailed by, contradicts, or is neutral to the premise, supporting the task of recognizing textual entailment. We will also include diagrams showing the relations between the components in our text generation framework and provide their implementation details in the appendix.
>
> Quality of adversaries: Yes, generating impressive natural adversaries against more accurate classifiers is difficult, since they require much more substantial changes to the original inputs and a more accurate representation of the data manifold than the current GANs are able to encode. But in essence, we utilize this exact phenomena to evaluate the accuracy and robustness of black-box classifiers qualitatively and quantitatively as shown in experiments, and hope to continue improving our approach to generate even better examples for such classifiers.
>
> Search algorithm: We have an improved search algorithm based on a coarse-to-fine idea that iteratively shrinks the upper bound of \Delta z. We will include this modification that results in much more efficient generation of samples in the revision (more details in the response to Reviewer 1).

---

### Official Review · AnonReviewer3 · 2017-11-27
**The authors present an interesting research problem of generating adversarial examples to show differences in predictions in black-box classifiers. However, I feel the novelty of the perturbation idea in semantic space is questionable and author needs to highlight the significance in a more explicit way.**

**Rating:** 6
**Confidence:** 3

**Review:**

Quality: Although the research problem is an interesting direction the quality of the work is not of a high standard. My main conservation is that the idea of perturbation in semantic latent space has not been described in an explicit way. How different it will be compared to a perturbation in an input space?

Clarity: The use of the term "adversarial" is not quite clear in the context as in many of those example classification problems the perturbation completely changes the class label (e.g. from "church" to "tower" or vice-versa)

Originality: The generation of adversarial examples in black-box classifiers has been looked in GAN literature as well and gradient based perturbations are studied too. What is the main benefit of the proposed mechanism compared to the existing ones?

Significance: The research problem is indeed a significant one as it is very important to understand the robustness of the modern machine learning methods by exposing them to adversarial scenarios where they might fail.

pros:
(a) An interesting problem to evaluate the robustness of black-box classifier systems
(b) generating adversarial examples for image classification as well as text analysis.
(c) exploiting the recent developments in GAN literature to build the framework forge generating adversarial examples.

cons:
(a) The proposed search algorithm in the semantic latent space could be computationally intensive. any remedy for this problem?
(b) Searching in the latent space z could be strongly dependent on the matching inverter $I_\gamma(.)$. any comment on this?
(c) The application of the search algorithm in case of imbalanced classes could be something that require further investigation.

---

> ### Author Response · Authors · 2017-12-15
> **Response to Reviewer 3**
>
> Thanks for the comments.
>
> Input perturbations vs. latent perturbations: We demonstrate an illustrative example in Figure 1 (d, e) showing the differences compared to perturbations in input space in Figure 1 (b, c). There are more FGSM examples provided in Table 1 showing the advantages of our approach. Moreover, approaches that add noise directly to the input are not applicable to complex data such as text because of the discrete nature of the domain. Adding imperceivable changes to the sentences is impossible, and perturbations often result in sentences that are not grammatical. Our framework can generate grammatical sentences that are meaningfully similar to the input by searching in the latent semantic space. There are also examples in Section 3.2 and appendix showing this advantage of our approach.
>
> Related work: To the best of our knowledge, there is no existing work on generating natural adversaries against black-box classifiers utilizing GANs. Other attack methods, none of which utilize GANs, either have access to the gradients of white-box classifiers, or train substitution models mimicking the target classifiers to attack. Further, these methods still add perturbations in input space, while our approach attacks target black-box classifiers directly and searches in the latent semantic space, generating natural adversaries that are legible/grammatical, meaningfully similar to the input, and helpful to interpret and evaluate the black-box classifiers, as demonstrated in our results. Please point us to the GAN literature that generates adversaries against black-box classifiers as mentioned in the review, and we will be happy to compare against them.
>
> Using the term "adversarial": Yes, there is an implicit assumption that the generated samples are within the same class if the added perturbations are small enough, and the generated samples look as if they belong to different classes when the perturbations are large. However, note that it is also the case for FGSM and other such approaches: when their \epsilon is small, the noise is imperceivable; but with a large \epsilon, one often finds noisy instances that might be in a different class (see Table 1, digit 8 for an example). While we do observe this behavior in some cases, the corresponding classifiers require much more substantial changes to the input and that is why we utilize our approach to evaluate black-box classifier. We will clarify this in the revision of the paper.
>
> Matching inverter: The generator/inverter in our approach work in similar way as the decoder/encoder in autoencoders. It is true that the quality of generated samples depends on these two components together. In Section 6, we mention that the fine-tuning of the latent vector produced by the inverter can further refine the generated adversarial examples, indicating that more powerful inverters are promising future directions of current work.
>
> Search algorithm: Gradient-based search methods such as FGSM are not applicable to our setup because of black-box classifiers and discrete domain application. We have an improved search algorithm by using a coarse-to-fine strategy that we will include in the revision (see our reply to Reviewer 1 for more details).

---

> > ### Comment · AnonReviewer3 · 2018-01-12
> > **Most of my concerns have been addressed**
> >
> > Most of my concerns have been properly addressed. I agree with the author that use of GAN to generate adversarial examples in text analysis is indeed novel. The importance and the application of the proposed methodology has now been depicted clearly.
> >
> > However I still have two small issues- (1) The application of the search algorithm for imbalanced classes and (2) computational complexity of the search algorithm (The authors also mention this in the paper- "Our iterative stochastic search algorithm for identifying adversaries is computationally expensive since it is based on naive sampling and local-search" -how to improve it?)
> >
> > Hence, although I have raised the score from my previous review, I feel it is only marginally above acceptance threshold.

---

### Official Review · AnonReviewer1 · 2017-11-27
**An interesting paper which is marginally above acceptance threshold**

**Rating:** 6
**Confidence:** 3

**Review:**

The authors of the paper propose a framework to generate natural adversarial examples by searching adversaries in a latent space of dense and continuous data representation (instead of in the original input data space). The details of their proposed method are covered in Algorithm 1 on Page 12, where an additional GAN (generative adversarial network) I_{\gamma}, which can be regarded as the inverse function of the original GAN G_{\theta}, is trained to learn a map from the original input data space to the latent z-space. The authors empirically evaluate their method in both image and text domains and claim that the corresponding generated adversaries are natural (legible, grammatical, and semantically similar to the input).

Generally, I think that the paper is written well (except some issues listed at the end). The intuition of the proposed approach is clearly explained and it seems very reasonable to me.
My main concern, however, is in the current sampling-based search algorithm in the latent z-space, which the authors have already admitted in the paper. The efficiency of such a search method decreases very fast when the dimensions of the z-space increases. Furthermore, such an approximation solution based on the sampling may be not close to the original optimal solution z* in Equation (3). This makes me feel that there is large room to further advance the paper. Another concern is that the authors have not provided sufficient number of examples to show the advantages of their proposed method over the other method (such as FGSM) in generating the adversaries. The example in Table 1 is very good; but more examples (especially involving the quantitative comparison) are needed to demonstrate the claimed advantages. For example, could the authors add such a comparison in Human Evaluation in Section 4 to support the claim that the adversaries generated by their method are more natural?

Other issues are listed as follows:
(1). Could you explicitly specify the dimension of the latent z-space in each example in image and text domain in Section 3?
(2). In Tables 7 and 8, the human beings agree with the LeNet in >= 58% of cases. Could you still say that your generated “adversaries” leading to the wrong decision from LeNet? Are these really “adversaries”?
(3). How do you choose the parameter \lambda in Equation (2)?

---

> ### Author Response · Authors · 2017-12-15
> **Response to Reviewer 1**
>
> Thank you for the comments.
>
> Search algorithm: Gradient-based search methods such as FGSM are not applicable to our setup because of black-box classifiers and applications with discrete domains. We have an improved version of the search algorithm that uses a coarse-to-fine strategy to iteratively minimize the upper-bound of \Delta z based on fewer samples, and then performs finer search in the restricted range recursively. We observe around 4 times speedup in practice and will include more details in the revision.
>
> Comparison: It is difficult to compare against FGSM quantitatively regarding how "natural" the adversaries are, but we will include more examples in the revision. On one hand, FGSM can add such a small magnitude noise that our eyes do not perceive. On the other hand, the noise added by FGSM, when amplified, looks random without any interpretable meaning to us. It is also worth mentioning that users found ~80% of our generated sentences natural (legible/grammatical), a domain for which FGSM cannot be applied at all.
>
> Details: The dimension of latent z vector for MNIST, LSUN, and SNLI are 64, 128, and 300 correspondingly. And we choose \lambda = 10 to emphasize the reconstruction error in latent space, after trying out different values and inspecting generated samples. We will include these details in the revision.

---

### Public Comment · (anonymous) · 2017-11-06
**Adversarial example belonging to a different class?**

Interesting research direction. However, considering Table 2, I was wondering if we take an image of "tower" and semantically change it to an image of "church," then how do we expect the classifier to classify it as "tower"? In essence, the adversarial example must belong to the same class as the original image, otherwise one can completely replace the original image with a new one.

---

> ### Author Response · Authors · 2017-11-07
> **Reply to "Adversarial example belonging to a different class?"**
>
> We are glad that the commenter finds the idea interesting. Natural adversarial examples are defined differently here from the conventional adversaries, where one is searching for minimal adversarial change to the input directly. Our objective is to find the minimal amount of semantic change to the input that results in different prediction in order to interpret the decision behavior of the classifier. Indeed, while the change in semantic space may sometimes be sufficiently substantial to make the generated sample actually end up in a different class, the sample is still generated from the minimal semantic change (not just some random sample of a different class), and the way in which it differs from the original input can provide useful insights into the classifier.

---

> > ### Public Comment · (anonymous) · 2017-11-10
> > **Adversarial and GAN Literature**
> >
> > I asked the question because you have used the term "adversarial" and my point was, if your method changes both the image and the label, it may not be suitable to call the modified image as adversarial.
> >
> > Moreover, I think the concept of slowly transiting from images of one class to another has been widely studied and examined in GAN literature.

---

### Author Response · Authors · 2017-12-31
**List of changes in the revision**

Thanks for all the reviews. We have submitted a revision with changes listed below:

- Page 3: added more efficient Algorithm 2 of hybrid shrinking search (with pseudocode in the appendix on Page 15); clarified how we choose hyper-parameter \lambda.
- Page 4 & 5: included dimensions of latent z used, and more details about SNLI dataset.
- Page 7: updated results (Table 6 and Figure 3) based on the new algorithm.
- Page 8: added human evaluation results supporting that our adversaries are more natural than those generated by FGSM.
- Page 9: clarified the common assumption that adversaries are within the same class if the added perturbations are small enough, and how we utilize it to evaluate black-box classifiers.
- Minor: corrected a few typos and wording issues.
- Appendix: included architecture diagrams and implementation details.

---

### Decision · Program_Chairs · 2018-01-29
**ICLR 2018 Conference Acceptance Decision**

**Decision:**

Accept (Poster)

**Comment:**

The paper proposes a method to generate adversaries close to the (training) data manifold using GANs rather than arbitrary adversaries. They show the effectiveness of their method in terms of human evaluation and success in fooling a deep network. The reviewers feel that this paper is for the most part well-written and the contribution just about makes the mark.